# Keeping it Simple – Computational Resources in Deep Generative versus Traditional Methods for Synthetic Tabular Data Generation in Healthcare

Anonymous Full Paper
Submission 21

## Abstract

Synthetic data has emerged as a solution to address data access challenges in healthcare, particularly for accelerating AI tool development. Deep generative methods, including generative adversarial networks, variational autoencoders, and diffusion models, have gained prominence for creating realistic and representative synthetic datasets with low re-identification risk. However, while sustainability of future computational needs is a growing topic, computational needs are often overlooked when documenting benchmarking of synthetic data generators.

This study compares computational resources needed using traditional and deep generative methods for generating a synthetic breast cancer dataset, relative to differences in statistical similarity between the training dataset and the synthetic dataset.

The findings reveal that while quality performance within this experiment is comparable, the deep generative methods consume significantly more resources, necessitating High Performance Computing resources. We recommend researchers will increasingly include computational resources as a parameter when benchmarking methods, to build a bigger canvas of literature to guide the method choice.

## 1 Introduction

There is a growing need for access to high quality data to develop, train or test new AI driven tools ([1]). Synthetic data is seen as a method to overcome the issue of access to sensitive healthcare data ([2]). Some synthetization methods demand high computational resources, making them less available for mainstream use and challenging its future sustainability. Deep generative methods like generative adversarial networks (GANs), variational autoencoders (VAEs) and diffusion models (DMs) have gained traction in the literature as preferred methods to producing high quality synthetic data. With an initial focus on image data, several methods have been tailored to better fit tabular data which is the predominant format for Electronic Health Record (EHR) data.

Although the methods are often reported to create realistic and representative datasets with minimal risk of re-identification, high computational complexity leads to resource requirements concerns ([3]) and should be a consideration for choice of method. Still, for synthetic tabular data in healthcare computational resource needs are rarely addressed in generation method evaluations ([4]).

To guide the practical use of these tools, this study provides an experiment comparing traditional statistical methods and deep generative methods investigating the following questions:

Q1: What are the practical resource requirements for running the pipelines of deep generative methods for generating tabular data compared to traditional generators, and

Q2: How well does deep generative methods perform for generating tabular data compared to traditional generators in terms of statistical similarity?

This experiment benchmarks a traditional method for synthesizing tabular data against deep generative methods, comparing performance in terms of computational resources and statistical similarity to the training data. The experiment on a synthetic breast cancer dataset shows that a traditional generation method – the GaussianCopula - did not require High Performance Computing (HPC) infrastructure yet performed similarly to the resource intensive methods in terms of an average statistical similarity score. This result shows there are situations when the simple methods can be as adequate for the task as the compute heavy methods, making tabular data synthetization available to a broader public in the healthcare sector outside of the academic community.

## 2 Related Work

Deep generative models seem to be gaining popularity in the literature in the field of tabular healthcare data ([5], [6]). Although it is widely acknowledged that these are more resource intensive than the traditional methods, few have documented the actual difference between the two classes of approaches. [7] point to specific models being developed to fit tabular data like HealthGAN ([8]), MedGAN ([9]), and CTGAN ([10]). A recent review by [4] on syn-

thetic tabular data for healthcare showed that approximately 1/3 of the articles benchmarked deep generative methods against traditional methods like CART ([11]), GaussianCopula ([12]), and Bayesian networks ([13]). Four of these articles mentioned preprocessing needs or computational resources in their evaluation, without necessarily reporting the metrics. None of the articles concluded on computational needs for specific types of generation methods ([4]). There seems to be a need for a greater focus on using sustainable computing or carbon footprint as a performance indicator in the literature.

# 3 Methods

To compare the performance of the deep generative models on tabular data (Q2), the experiment was run with a GaussianCopula model as a benchmark to the Conditional Tabular Generative Adversarial Network (CTGAN) and Tabular Variational Autoencoder (TVAE). An experiment was run both on a laptop and in a virtual machine (VM) in Microsoft Azure with NVIDIA GPU HPC facilities on the Veracity platform ([14]). The Synthetic Data Vault (SDV) pipeline ([15]) for evaluation metrics for the models were compared to answer Q2, and practical performance measures (size and runtime) was gathered for answering Q1.

## 3.1 Dataset

The experiment was run using a synthetically generated dataset from the Dutch cancer registry that can be accessed by applying at ([16]), see an extract in Figure A.1 in the Appendix. The dataset is a cohort of 60.000 hypothetical breast cancer patients and has been created with the intention of showcasing what real healthcare data looks like, inheriting both the statistical patterns but also the typical traits of real-world healthcare data like missing values.

Each patient has one row with 47 columns of information about the episode, including measurements of tumor size and location, treatments etc. The table has 60.000 rows x 46 columns (features). Some variables are discrete, and others are continuous. Further details on the data are enclosed in Appendix.

As the dataset has been produced to optimize structural similarity while preserving privacy, this data cannot be expected to be clinically valid and should only be used for methodological experiments. However, the dataset remains useful for comparing the computational requirements of different generators.

## 3.2 Data preparation

Assuring quality for synthetic data generation starts with the training data [4]. Initial cleaning of the dataset revealed that 10 rows contained missing fields. For the simplicity of this exercise, the rows of patients with missing data were removed. The resulting number of rows left for analysis was 59 990. In a real-life experiment, this may not be an optimal approach as missing data fields is an inherent characteristic of healthcare data and the lack of data may harbor clinical information.

The pipeline was tested both with and without meta data definition. For the meta data definition, it was assumed there is only one tumor per row, even though a patient can have several tumors. This assumption makes both variables patient and episode IDs key-nkr and key-eid primary keys.

## 3.3 Choice of synthetic data generation models

To simplify the reproducibility of our results and ensure consistency in the implementation of data generators, we opted for using the well-documented ([17]) and maintained open-source python library for synthetic data generation, the SDV library ([15]). The SDV pipeline was chosen as it was the most cited open-source pipeline available at the time of the experiment. This library provides a reliable and standardized framework for synthetic data generation, minimizing the discrepancies that might arise from using different tools or custom implementations. The SDV library's comprehensive documentation and active community support further facilitate the reproducibility and validation of experimental outcomes. Additionally, its origins as a project at MIT lend it academic credibility in the field of synthetic data generation.

A traditional statistical model – the GaussianCopula ([18]) was used to compare the performance of two deep generative models, the TVAE ([19]) and CTGAN ([20]) for the tabular healthcare data.

The GaussianCopulaSynthesizer ([21]) is based on copula functions. A copula in mathematical terms maps the marginal distribution of a variable to the normal distribution through the probability integral transform. This mathematical function allows a description of the joint distribution of multiple variables, analyzing the dependencies between their marginal distributions ([18]).

The TVAE (Tabular Variational Autoencoder) is an adaptation of the variational autoencoder architecture specifically designed for tabular data. It uses an encoder-decoder structure where the encoder compresses the input data into a latent space representation, and the decoder reconstructs the data from this latent space. The TVAE is particularly effective at capturing complex relationships between

variables in tabular datasets. The CTGAN (Conditional Tabular Generative Adversarial Network) is a GAN-based model tailored for tabular data. It uses a generator to create synthetic samples and a discriminator to distinguish between real and synthetic data. The "conditional" aspect allows it to handle both continuous and discrete columns effectively, making it well-suited for heterogeneous tabular data often found in healthcare datasets. Both the TVAE and CTGAN models were presented at the NeurIPS 2019 conference in the paper titled "Modeling Tabular data using Conditional GAN" ([10]).

### 3.4 Setting up the infrastructure

The experiments were conducted on two platforms: (1) a Dell laptop with an Intel i7 CPU and embedded Intel GPU (without NVIDIA GPU acceleration), and (2) a virtual machine on the Microsoft Azure-based platform DNV Veracity, equipped with an NVIDIA M60 GPU (including CUDA acceleration). The pipelines for TVAE, CTGAN, and Gaussian-Copula were deployed on both platforms to compare performance.

### 3.5 Running the experiment

The original data was split in 20/80. Due to resource consumption, 20 percent (12K patients) were used to train the model while the remaining 80 percent was reserved for evaluating model performance. This split allows us to assess not only how well the synthetic data captures the training data patterns but also how well these patterns generalize to unseen data. On the virtual machine, all three models were trained and tested using two distinct evaluation approaches: (1) unsupervised evaluation (comparing the statistical properties of synthetic data directly with the training data to assess structural similarity); and (2) supervised evaluation (training predictive models on both real and synthetic data, then evaluating their performance on the test set to assess the utility of the synthetic data).

The training was done first without defining meta data – describing whether a column (feature) is numerical or categorical. An additional test was done using only TVAE and the GaussianCopula after manually defining this meta data, to evaluate the effect. To facilitate the reproducibility of our experiments, we used the default hyperparameters as defined in the source code for the corresponding generators ([22], [23], [24]).

For the laptop experiment, a smaller amount of training data was used (10 percent of the data). The deep generative models were trained and tested to explore resource consumption. The CTGAN had to be aborted due to resource overload.

### 3.6 Quality evaluation

Our evaluation framework employed both unsupervised and supervised approaches to comprehensively assess synthetic data quality. The SDV Evaluation Metrics Library from the original Synthetic Data Vault Project ([15]) was used to evaluate the synthetic data generated from each model. The evaluation included SDV's quality report and diagnostics report. The quality report ([25]) presented weighted scores on column shapes and column pair trends. The column pair trends describe structural similarity between the synthetic data and the test data; how they vary in relation to each other, for example the correlation. The higher the score, the more the trends are alike ([15]).

The diagnostic report ([26]) presented two similarity scores that compare synthetic data with the test data, and one score on privacy risk: coverage and boundaries (similarity) and overfitting/copying of test data (privacy risk) ([27]).

Coverage means how well the synthetic data covers the categories present in the real data. Boundaries is a measure of how well the synthetic data follows the min/max boundaries set by the real data. The score is between 0 and 1, and the higher the score the better. Overfitting is evaluated by a measure of how many rows are copies of the original data.

Using an automated pipeline for quality evaluation is easy to implement and use but may not show all quality dimensions that should be investigated if the synthetic data is to be used in a safety critical clinical context ([4]). While average scores are practial for benchmarks, they can obscure deviating performance for specific features, and diagnostic exercises was performed to evaluate featurewise similarity in addition to the scores. In this pipeline, there was no bias or fairness metrics, no clinical usability, and only one privacy measure. For the context of this experiment, this was deemed sufficient.

## 4 Results

There were significant differences in the resource needs for running the generation pipelines, and only minor differences in the measured average statistical similarity between the generators.

The computational needs for the generative methods were significantly higher than the traditional method, to the point where certain parts of the experiment were not feasible to complete. When meta data was defined, the GaussianCopula outperformed the two other models in all similarity scores. When meta data was not defined, the performance of GaussianCopula dropped to below that of the deep generative method TVAE, the TVAE performance was more robust to the lack of meta data.

The results section is divided into answering the

**Table 1.** Practical performance of the different models, training time and model size. Only the experiment without meta data was run on the laptop.
*\* The CTGAN laptop experiment was manually aborted.*
*\*\* The CTGAN HPC experiment without meta data was not completed due to CUDA out of memory errors.*

| Generation method | Time laptop | Time Azure | Model size |
| --- | --- | --- | --- |
| **WITHOUT META DATA** | | | |
| GaussianCopula | NA | 17 sec | 1.5 Mb |
| CTGAN | more than 420.000 sec* | NA** | 7825 Mb |
| TVAE | 54.000 sec | 1200 sec | 7.5 Mb |
| **WITH META DATA** | | | |
| GaussianCopula | NA | 4 sec | 0.5 Mb |
| CTGAN | NA | 480 sec | 33 Mb |
| TVAE | NA | 120 sec | 0.5 Mb |

**Table 2.** Quality evaluation scores of the models run with and without manually defined meta data. Scores are between 0 and 1, higher score is better quality. Description of the quality parameters in section 3.6. Quality evaluation and details can be found in the [15] library. (See larger version and data labels dictionary in appendix.)
*\*\* The CTGAN HPC experiment without meta data was not completed due to CUDA out of memory errors.*

| Metrics | GaussianCopula | CTGAN | TVAE |
| --- | --- | --- | --- |
| **WITH META DATA** | | | |
| **Quality score** | | | |
| Column shapes | 0.9247 | 0.8831 | 0.8813 |
| Column pair trends | 0.8711 | 0.8211 | 0.8158 |
| **Diagnostics** | | | |
| Coverage | 0.97 | 1.0 | 0.93 |
| Copies | 1.0 | 1.0 | 1.0 |
| Boundaries | 1.0 | 1.0 | 1.0 |
| **WITHOUT META DATA** | | | |
| **Quality score** | | | |
| Column shapes | 0.76 | NA** | 0.81 |
| Column pair trends | 0.61 | NA** | 0.72 |
| **Diagnostics** | | | |
| Coverage | 0.92 | NA** | 0.76 |
| Copies | 1.0 | NA** | 1.0 |
| Boundaries | 1.0 | NA** | 1.0 |

two questions, respectively.

## 4.1 Pipeline requirements for processing capacity (Q1)

The models were trained on 20 percent of the cleaned original dataset (11.998 records) on the virtual machine, both with and without defining meta data (numerical vs categorical values). Table 1 shows results of training time and storage needs for the models. The training time was notably shorter for the traditional GaussianCopula model compared to the deep generative models with a factor ranging from 30 to 565 times faster. Without defined meta data, and running on the azure server, the traditional GaussianCopula model used 17 seconds and the TVAE used 1200 seconds. Notably, the CTGAN evaluation in the "without meta data" scenario could not be completed due to CUDA out of memory errors on our GPU setup, highlighting the substantial memory requirements of this model when processing unstructured data. With meta data the training was faster for all models, rendering the relative differences smaller but with a similar distribution. Storage was notably smaller when the meta data was defined.

The laptop setup proved impractical for the deep generative methods. While the TVAE generator could run, it took an excessively long time. The CTGAN was stopped after two days due to its prolonged runtime. These results highlight the necessity of High Performance Computing (HPC) resources for deep generative methods when working with tabular data of this scale.

## 4.2 Similarity benchmark of generator performance on tabular data (Q2)

Table 2 shows results from the similarity evaluation on the synthetic data run on models with and without defined meta data (as detailed in meta data schema of SDV developer guide 13). With meta data, the TVAE performs adequately but not as well as the simple statistical approach (GaussianCopula) on the tabular data, with slightly lower scores for column shapes, column pair trends and coverage. The TVAE shows slightly lower quality scores than the CTGAN and lower coverage. The table values for metrics in the experiment with manually defined meta data have four digits to emphasize the difference between CTGAN and TVAE.

Analysing the coverage pr feature, the column shapes showed a more consistent performance with the GaussianCopula with the average score of 0.97 versus the TVAE average score of 0.93. The graphs for column coverage (see figure 1) show a difference in the performance for numerical versus (dark blue columns) categorical values (light blue columns). The TVAE performs better than the GaussianCopula on numerical values, while the GaussianCopula performs better for the categorical values. The same trend was seen on column shapes.

The TVAE performed noticeably worse for one specific variable (tumor morphology). The distribution of this variable was quite particular and difficult to capture for a generator, with ductal carcinoma (code 8500) being noticeably larger than the other groups with 44.426 instances, Lobular carcinoma (code 8520) with 7245 instances and the residual 28 groups having a count below 1000 instances (224-889 instances). The distribution of the original data for this feature is shown in Figure A.4.

When meta data was not manually defined, the TVAE showed better overall similarity performance scores than the GaussianCopula, showcasing the potential of deep generative models compared to tra-

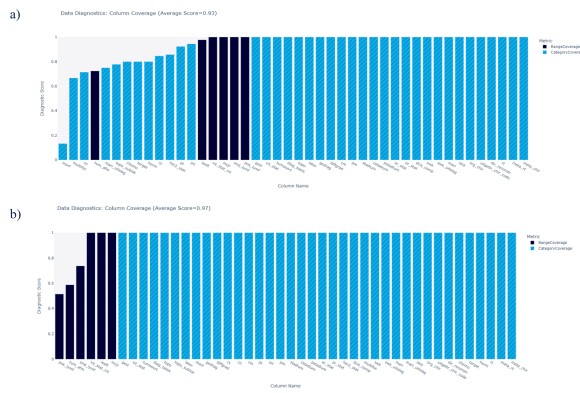

**Figure 1.** The figure shows a detailing of the average column coverage score provided in Table 2 in the experiment when the meta data was defined, comparing a) the TVAE results and b) the GaussianCopula results. The dark blue RangeCoverage is used for numerical values and the light blue CategoryCoverage is for categorical data. See the appendix for a bigger version of the image.

ditional models, for learning the underlying patterns of a dataset.

## 5  Discussion

All pipelines were faster when meta data was manually defined. The statistical GaussianCopula pipeline was the fastest and smallest in terms of storage need. When it is necessary to work with the computationally demanding deep generative pipelines, it could be beneficial to use the statistical model to iterate on your experiments and see that your meta data is correct before running the more time-consuming generators. It was possible but not practical to run the TVAE generator on a laptop since the size of the seed dataset had to be limited and it had an impractically long runtime (15 hours). The resource needs for the CTGAN in this experiment confirms the potential extent of computational requirements for tabular data generation with deep generative models, even in the HPC setting. For the community to move forward in a sustainable way, it would be beneficial to choose traditional and less compute intensive methods when quality needs and performance requirements allow it.

When choosing a generation method, one should consider all relevant quality dimensions– similarity, usability (clinical and other), privacy, bias and fairness and carbon footprint ([4]) and prioritize them according to the intended downstream use of the dataset. While investigating all perspectives is particularly important in potential high-impact areas like healthcare, fairness was excluded in this experiment due to the dataset composition (one gender and homogeneous population, lacking minority groups).

The Gaussian Copula had the highest average quality score when meta data was defined, while the TVAE showed better consistency in performance independently of predefined meta data. The detailed findings of the column coverage (figure 1 ) shows that for certain parameters (tumor morphology) the TVAE showed poor column coverage. This can be explained by the disproportionate distribution of this variable in the training data. Gaussian Copula seemed to perform noticeably worse on numerical data compared to the TVAE, but as there were few numerical values in the dataset this did not affect the average scores significantly.

An average score such as the SDV quality evaluation score used in this paper is useful for benchmarking generators on a generic level, but in choosing the relevant quality parameters for a specific clinical case, one must consider the cohort and what clinical desiderata are relevant for these according to the intended use.

A conclusion on quality performance and choice of an optimal generation model cannot be generic and must always be adjusted to the specific data and clinical case at hand.

**Limitations**  This paper is a synthetic experiment with no downstream use, and therefore no considerations for specific parameters have been discussed. The training dataset used in the experiment is a synthetically generated dataset. Since the real data is not accessible, the actual objective quality compared to reality cannot be determined. Creating a synthetic dataset based on a synthetic dataset can only be seen as a practical or methodological test. An important limitation of this study is the reliance on a single library (SDV) for implementation of the synthetic data generation models. While SDV is well-documented and widely cited, using a single library may limit the generalizability of our findings. Future studies should consider implementing models using multiple libraries to provide a more comprehensive comparison. There are also other open libraries available like the recently published SynthCity at Cambridge ([28]), and new evaluation metrics are being proposed in the literature. The SDV pipeline could preferably be expanded by additional quality metrics ([29]). In the experiment, we relied on two out-of-the-box reports natively provided by the SDV library: "The Diagnostic Report" and "The Quality Report". According to the library developers, these two reports provide scores for an aggregated and overall comparison of original and synthetic datasets. While this approach offers a simplified version of reporting, focusing on computational resources and their comparisons, we acknowledge that it lacks more rigorous statistical testing. In the experiment, the standard SDV pipeline metrics were used as these are perceived as commonly used for statistical sim-

ilarity. It is unclear whether this score properly rewards the desired model behaviour, resulting in uncertainty around the results. The evaluation metrics used in this case will not evaluate the usability of the data and the clinical logic like the realism of the TNM classification of tumors in the different cases or in downstream performance. Fairness metrics are not included, and only a simple similarity metric of statistical closeness is used for measuring privacy. The study is limited to only three generation methods. In the future, we plan to investigate newer and potentially more effective models such as diffusion and large language models.

To conclude on the actual clinical utility of the synthetic data produced by the pipelines, this experiment should be recreated on a real dataset and for a defined downstream application. Other quality metrics and mechanisms should also be added, to evaluate the clinical usability and logic of the resulting synthetic data.

To better guide practitioners in making sustainable choices, computational resources should be included as a reporting parameter for benchmarking of methods. To ensure a focus on sustainable computing, metrics of training time and model size could be translated into a more comparable metric like carbon footprint.

**Further work** This experiment covers one data modality - tabular data - with synthetic national cancer registry type data on breast cancer, and using two deep generative methods and one traditional method. The generators used were considered most common at the time of the experiment and available in a public pipeline. To investigate the robustness of the conclusions, there is a need for a wide range of benchmarks on a range of different datasets, investigating other types of both deep and traditional generators and for differing use cases and data modalities, including more thorough hyperparameter exploration. Future studies could consider implementing models from multiple libraries to provide a more comprehensive comparison. For instance, the recently published SynthCity library developed at Cambridge University ([28]) could be used alongside SDV to broaden the scope of the analysis and validate findings across different implementations. In addition, future research should investigate the differences in quality scores for a broader range of evaluation criteria covering topics like fairness, privacy and other representativity (similarity and usability) metrics.

## 6   Data availability

The synthetic data from the Dutch breast-cancer cohort is available at the website of the Cancer Registry of Netherland ([16]).

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

# A  Figures and images

This appendix includes larger images of the figures in the article and an additional data dictionary from the IKNL dataset.

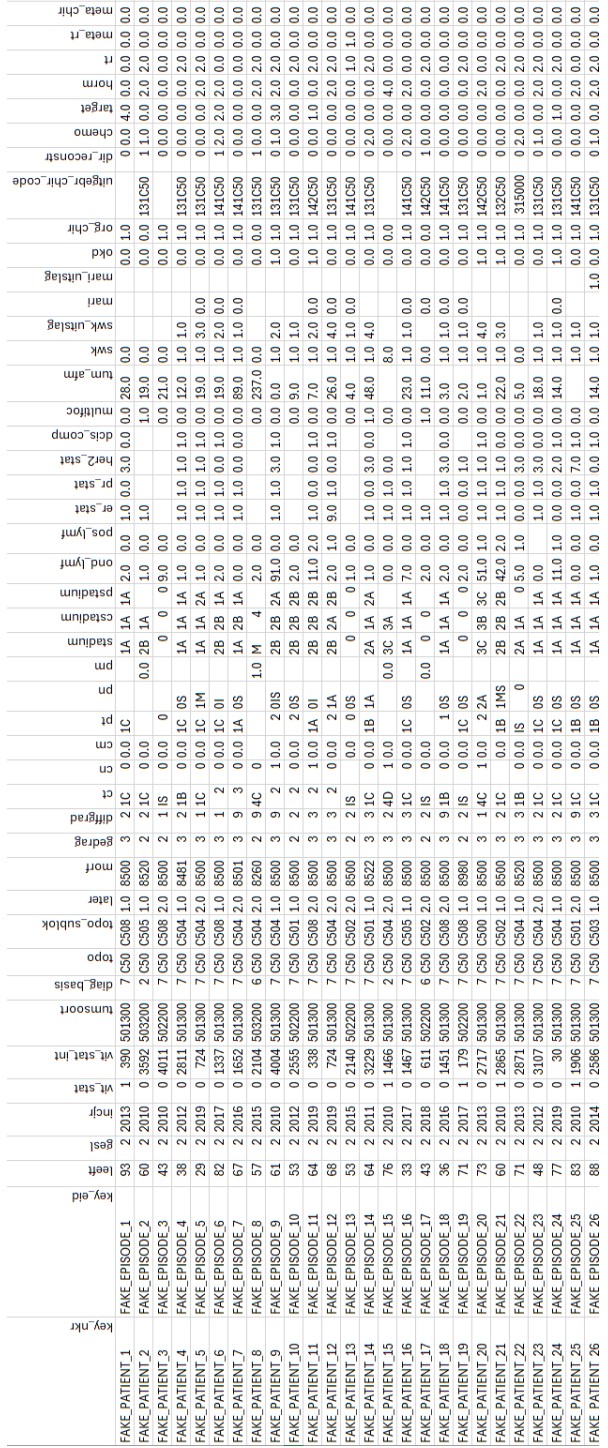

**Figure A.1.** Snip of the raw data from the IKNL dataset

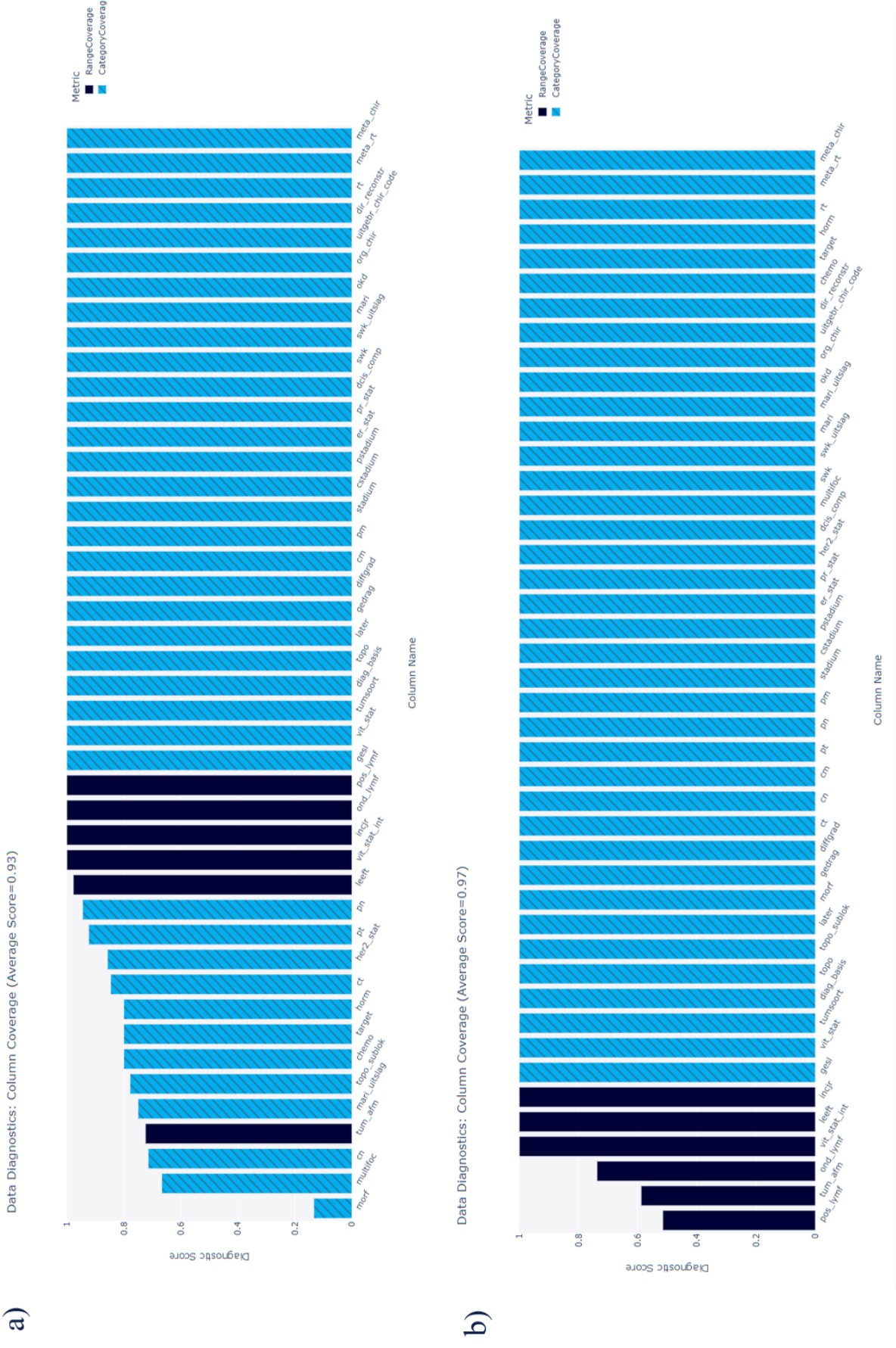

**Figure A.2.** Detailing of the average column coverage score when the meta data was defined, comparing a) TVAE results and b) GaussianCopula results. The dark blue RangeCoverage is for numerical values and the light blue CategoryCoverage is for categorical data.

| Codelijst | Code | ENGLISH | Omschrijving |
|---|---|---|---|
| behprepost | 0 | No | Nee |
| behprepost | 1 | Yes, only preop | Ja, alleen prechirurgisch |
| behprepost | 2 | Ja only post op | Ja, alleen postchirurgisch |
| behprepost | 3 | Ja, pre and post op | Ja, pre- en postchirurgisch |
| behprepost | 4 | Yes, no suregery | Ja, geen chirurgie |
| diag_basis | 1 | Only clinical examination (anamnese eand physical) | Alleen klinisch onderzoek (anamnese en lichamelijk onderzoek) |
| diag_basis | 2 | Clinical-diagnostic examinations, exploratory surgery, or autopsy (without microscopic confirmation) | Klinisch-diagnostische onderzoeken, exploratieve chirurgie of obductie(zonder microscopische bevestiging) |
| diag_basis | 4 | Specific biochemical and/or immunological laboratory tests | Specifieke biochemische en/of immunologische laboratoriumonderzoeken |
| diag_basis | 5 | Hematological or cytological confirmation of the primary tumor or metastases, or there is microscopic confirmation but it's unclear whether it involves cytology or histology | Hematologische of cytologische bevestiging van de primaire tumor of metastasen, of er is microscopische bevestiging maar het is onduidelijk of dit cytologie of histologie betref |
| diag_basis | 6 | Histological confirmation of metastases only, including confirmation in autopsy | Histologische bevestiging uitsluitend van metastase(n), inclusief bevestiging bij obductie |
| diag_basis | 7 | Histological confirmation of the primary tumor, or unclear whether histological confirmation refers to the primary tumor or metastasis, and/or autopsy (with histological confirmation) | Histologische bevestiging van de primaire tumor, of onduidelijk of histologische bevestiging de primaire tumor of een metastase betreft. En/of obductie (met histologische bevestiging) |
| diffgrad | 1 | Well, highly differentiated / low-grade | Goed, hoog gedifferentieerd / laaggradig |
| diffgrad | 2 | Moderately differentiated / intermediate | Matig gedifferentieerd / intermediair |
| diffgrad | 3 | Poorly differentiated / high-grade | Slecht, weinig gedifferentieerd / hooggradig |
| diffgrad | 4 | Undifferentiated / anaplastic | Ongedifferentieerd / anaplastisch |
| diffgrad | 9 | Unknown / n/a / not determined | Onbekend / n.v.t. / niet bepaald |
| gedrag | 2 | In situ | In situ |
| gedrag | 3 | Malignant | Maligne |
| gesl | 1 | Male | Man |
| gesl | 2 | Female | Vrouw |
| her2_stat | 0 | 0, negative | 0, negatief |
| her2_stat | 1 | 1+, negative | 1+, negatief |
| her2_stat | 2 | 2+, unclear | 2+, onduidelijk |
| her2_stat | 3 | 3+, positive | 3+, positief |
| her2_stat | 4 | Not determined | Niet bepaald |
| her2_stat | 7 | Not determined | Niet bepaald |
| her2_stat | 9 | Unable to assess, unknown | Niet te beoordelen, onbekend |
| hr_stat | 0 | Negative | Negatief |
| hr_stat | 1 | Positive | Positief |
| hr_stat | 9 | Unable to assess, unknown | Niet te beoordelen, onbekend |
| mari_uitslag | 1 | Mari-lymph node negative | Mari-klier negatief |
| mari_uitslag | 2 | ITC (≤ 0.2 mm) | ITC (<= 0.2 mm) |
| mari_uitslag | 3 | Micrometastasis (> 0.2 - ≤ 2 mm) | Micrometastase (> 0.2 - <= 2 mm) |
| mari_uitslag | 4 | Mari-lymph node positive | Mari-klier positief |
| mari_uitslag | 5 | Mari-lymph node not removed | Mari-klier niet weggehaald |
| mari_uitslag | 9 | Result unknown | Uitslag onbekend |
| neeja | 0 | No | Nee |
| neeja | 1 | Yes | Ja |
| neejaonb | 0 | No | Nee |
| neejaonb | 1 | Yes | Ja |
| neejaonb | 9 | Unknown | Onbekend |
| swk | 0 | No | Nee |
| swk | 1 | Yes | Ja |
| swk | 8 | Not registered in region | Niet geregistreerd in regio |
| tumsoort | 501300 | Invasive breast carcinoma | Invasief mammacarcinoom |
| tumsoort | 502200 | Ductal carcinoma in situ | Ductaal carcinoma in situ |
| tumsoort | 503200 | Lobular carcinoma in situ | Lobulair carcinoma in situ |
| vit_stat | 0 | Alive | In leven |
| vit_stat | 1 | Deceased | Overleden |
| swk_uitslag | 1 | Sentinel lymph node negative | Schildwachtklier negatief |
| swk_uitslag | 2 | ITC (≤ 0.2 mm) | ITC (≤ 0.2 mm) |
| swk_uitslag | 3 | Micrometastasis (> 0.2 - ≤ 2 mm) | Micrometastase (>0.2 - ≤2 mm) |
| swk_uitslag | 4 | Sentinel lymph node positive (> 2 mm) | Schildwachtklier positief (> 2 mm) |
| swk_uitslag | 9 | Sentinel lymph node not found | Schildwachtklier niet gevonden |
| topo | C50 | Breast | Borst |
| topo_sublok | C500 | Breast nipple/areola | Mamma tepel/tepelhof |
| topo_sublok | C501 | Breast central part | Mamma centraal deel |
| topo_sublok | C502 | Breast medial upper quadrant | Mamma mediaal bovenkwadrant |
| topo_sublok | C503 | Breast medial lower quadrant | Mamma mediaal onderkwadrant |
| topo_sublok | C504 | Breast lateral upper quadrant | Mamma lateraal bovenkwadrant |
| topo_sublok | C505 | Breast lateral lower quadrant | Mamma lateraal onderkwadrant |
| topo_sublok | C506 | Breast axillary tail | Mamma axillaire uitloper |
| topo_sublok | C508 | Breast overlapping | Mamma overlappend |
| topo_sublok | C509 | Breast NNO | Mamma NNO |
| later | 1 | Left | Links |
| later | 2 | Right | Rechts |
| later | X | Unknown | Onbekend |
| morfologie | 8000 | Neoplasm, NNO | Neoplasma, NNO |
| morfologie | 8001 | Malignant tumor cells | Maligne tumorcellen |
| morfologie | 8004 | Malignant tumor, spindle cell type | Maligne tumor, spoelceltype |
| morfologie | 8010 | Carcinoma, NNO | Carcinoom, NNO |
| morfologie | 8012 | Large cell carcinoma, NNO | Grootcellig carcinoom, NNO |
| morfologie | 8013 | Large cell neuroendocrine carcinoma | Grootcellig neuro-endocrien carcinoom |
| morfologie | 8020 | Undifferentiated carcinoma, NNO | Ongedifferentieerd carcinoom, NNO |
| morfologie | 8022 | Pleomorphic carcinoma | Pleiomorf carcinoom |
| morfologie | 8030 | Giant cell and spindle cell carcinoma | Reuscel- en spoelcelcarcinoom |
| morfologie | 8032 | Spindle cell carcinoma, NNO | Spoelcelcarcinoom, NNO |
| morfologie | 8033 | Pseudosarcomatous carcinoma | Pseudosarcomateus carcinoom |

| Codelijst | Code | ENGLISH | Omschrijving |
|---|---|---|---|
| morfologie | 8035 | Carcinoma with osteoclast-like giant cells | Carcinoom met osteoclastachtige reuscellen |
| morfologie | 8041 | Small cell carcinoma, NNO | Kleincellig carcinoom, NNO |
| morfologie | 8045 | Mixed small and large cell carcinoma | Gemengd klein- en grootcellig carcinoom |
| morfologie | 8046 | Non-small cell carcinoma | Niet-kleincellig carcinoom |
| morfologie | 8070 | Squamous cell carcinoma, NNO | Plaveiselcelcarcinoom, NNO |
| morfologie | 8071 | Keratinizing squamous cell carcinoma | Verhoornend plaveiselcelcarcinoom |
| morfologie | 8074 | Squamous cell carcinoma, spindle cell type | Plaveiselcelcarcinoom, spoelceltype |
| morfologie | 8082 | Lymphoepithelial carcinoma | Lymfo-epitheliaal carcinoom |
| morfologie | 8140 | Adenocarcinoma, NNO | Adenocarcinoom, NNO |
| morfologie | 8141 | Scirrhous adenocarcinoma | Scirreus adenocarcinoom |
| morfologie | 8145 | Adenocarcinoma, diffuse type | Adenocarcinoom, diffuus type |
| morfologie | 8200 | Adenoid cystic carcinoma | Adenoïd cysteus carcinoom |
| morfologie | 8201 | Cribriform carcinoma | Cribriform carcinoom |
| morfologie | 8211 | Tubular adenocarcinoma | Tubulair adenocarcinoom |
| morfologie | 8230 | Solid carcinoma, NNO | Solide carcinoom, NNO |
| morfologie | 8240 | Neuroendocrine tumor, NNO/grade 1 (carcinoid) | Neuro-endocriene tumor, NNO/graad 1 (carcinoid) |
| morfologie | 8244 | Mixed adenoneuroendocrine carcinoma (MANEC) | Gemengd adenoneuroendocrien carcinoom (MANEC) |
| morfologie | 8246 | Neuroendocrine carcinoma, NNO | Neuro-endocrien carcinoom, NNO |
| morfologie | 8249 | Neuroendocrine tumor, grade 2/3 (atypical carcinoid) | Neuro-endocriene tumor, graad 2/3 (atypisch carcinoid) |
| morfologie | 8255 | Adenocarcinoma with mixed subtypes | Adenocarcinoom met gemengde subtypes |
| morfologie | 8260 | Papillary adenocarcinoma, NNO | Papillair adenocarcinoom, NNO |
| morfologie | 8290 | Oxyphilic adenoma/carcinoma - Hurthle cell carcinoma | Oxyfiel adenoom/carcinoom - Hurthle-cel carcinoom |
| morfologie | 8310 | Clear cell adenocarcinoma, NNO | Heldercellig adenocarcinoom, NNO |
| morfologie | 8314 | Lipid-rich carcinoma | Lipidenrijk carcinoom |
| morfologie | 8315 | Glycogen-rich carcinoma | Glycogeenrijk carcinoom |
| morfologie | 8401 | Apocrine adenocarcinoma | Apocrien adenocarcinoom |
| morfologie | 8407 | Microcystic adnexal carcinoma / sclerosing sweat gland carcinoma | Microcysteus adnexcarcinoom/Scler. zweetkliercarcinoom |
| morfologie | 8410 | Sebaceous gland adenocarcinoma | Talgklieradenocarcinoom |
| morfologie | 8430 | Mucoepidermoid carcinoma | Muco-epidermoïd carcinoom |
| morfologie | 8441 | Serous cystadenocarcinoma, NNO | Sereus cystadenocarcinoom, NNO |
| morfologie | 8470 | Mucinous cystadenocarcinoma, NNO | Mucineus cystadenocarcinoom, NNO |
| morfologie | 8480 | Mucinous adenocarcinoma | Mucineus adenocarcinoom |
| morfologie | 8481 | Mucin-forming adenocarcinoma | Slijmvormend adenocarcinoom |
| morfologie | 8490 | Signet ring cell carcinoma / 'poorly cohesive' carcinoma | Zegelringcelcarcinoom / 'poorly cohesive' carcinoom |
| morfologie | 8500 | Ductal carcinoma, NNO | Ductaal carcinoom, NNO |
| morfologie | 8501 | Comedo carcinoma, NNO | Comedocarcinoom, NNO |
| morfologie | 8502 | Secretory carcinoma | Secretoir carcinoom |
| morfologie | 8503 | Intraductal papillary adenocarcinoma | Intraductaal papillair adenocarcinoom |
| morfologie | 8504 | Encapsulated (intracystic) papillary carcinoma | Omkapseld (intracysteus) papillair carcinoom |
| morfologie | 8507 | Intraductal micropapillary carcinoma | Intraductaal micropapillair carcinoom |
| morfologie | 8508 | Cystic hypersecretory carcinoma | Cystisch hypersecretoir carcinoom |
| morfologie | 8509 | Solid papillary carcinoma | Solide papillair carcinoom |
| morfologie | 8510 | Medullary carcinoma, NNO | Medullair carcinoom, NNO |
| morfologie | 8512 | Medullary carcinoma with lymphoid stroma | Medullair carcinoom met lymfoïd stroma |
| morfologie | 8513 | Atypical medullary carcinoma | Atypisch medullair carcinoom |
| morfologie | 8514 | Ductal carcinoma, desmoplastic type | Ductaal carcinoom, desmoplastisch type |
| morfologie | 8519 | Pleomorphic lobular carcinoma in situ | Pleiomorf lobulair carcinoma in situ |
| morfologie | 8520 | Lobular carcinoma, NNO | Lobulair carcinoom, NNO |
| morfologie | 8521 | Ductal carcinoma | Ductulair carcinoom |
| morfologie | 8522 | Ductal and lobular carcinoma | Ductaal en lobulair carcinoom |
| morfologie | 8523 | Ductal carcinoma, mixed with another carcinoma type | Ductaal carcinoom, gemengd met ander carcinoomtype |
| morfologie | 8524 | Lobular carcinoma mixed with another carcinoma type | Lobulair carcinoom gemengd met ander carcinoomtype |
| morfologie | 8530 | Inflammatory carcinoma | Inflammatoir carcinoom |
| morfologie | 8540 | Paget's disease of the breast | Morbus Paget van mamma |
| morfologie | 8541 | Paget's disease with infiltrating ductal carcinoma | Morbus Paget en infiltrerend ductaal carcinoom |
| morfologie | 8543 | Paget's disease with intraductal carcinoma (DCIS) | Morbus Paget en intraductaal carcinoom (DCIS) |
| morfologie | 8550 | Acinar cell carcinoma | Acinuscelcarcinoom |
| morfologie | 8560 | Adenosquamous carcinoma | Adenosquameus carcinoom |
| morfologie | 8562 | Epithelial-myoepithelial carcinoma | Epitheliaal-myoepithliaal carcinoom |
| morfologie | 8570 | Adenocarcinoma with squamous cell metaplasia | Adenocarcinoom met plaveiselcelmetaplasie |
| morfologie | 8571 | Adenocarcinoma with (chondroid) osseous metaplasia | Adenocarcinoom met (kraak)benige metaplasie |
| morfologie | 8572 | Adenocarcinoma with spindle cell metaplasia | Adenocarcinoom met spoelcelmetaplasie |
| morfologie | 8573 | Adenocarcinoma with apocrine metaplasia | Adenocarcinoom met apocriene metaplasie |
| morfologie | 8574 | Adenocarcinoma with neuroendocrine differentiation | Adenocarcinoom met neuro-endocriene differentiatie |
| morfologie | 8575 | Metaplastic carcinoma, NNO | Metaplastisch carcinoom, NNO |
| morfologie | 8980 | Carcinosarcoma, NNO | Carcinosarcoom, NNO |
| morfologie | 8982 | Myoepithelial carcinoma | Myo-epitheliaal carcinoom |
| morfologie | 8983 | Malignant adenomyoepithelioma | Maligne adenomyo-epithelioom |
| therapie | 100000 | Surgery NNO | Chirurgie nno |
| therapie | 120000 | Local tumor resection | Lokale tumorresectie |
| therapie | 130C50 | Breast-conserving surgery NNO | Borstsparende chirurgie nno |
| therapie | 131C50 | Lumpectomy (without sentinel lymph node biopsy) | Lumpectomie (zonder OKD) |
| therapie | 132C50 | Lumpectomy (with sentinel lymph node biopsy) | Lumpectomie (met OKD) |
| therapie | 140C50 | Non-breast-conserving surgery NNO | Niet-borstsparende chirurgie nno |
| therapie | 141C50 | Mastectomy (without sentinel lymph node biopsy) | Ablatio (zonder OKD) |
| therapie | 142C50 | Amputation (with sentinel lymph node biopsy) | Amputatie (met OKD) |
| therapie | 190000 | Resection for other indication (incidental finding) | Resectie voor andere indicatie (toevalsbevinding) |
| therapie | 315000 | Lymph node dissection of regional lymph node metastases | Lymfeklierdissectie regionale lymfekliermetastasen |
| therapie | 690100 | Surgical treatment abroad | Chirurgische behandeling in buitenland |

**Figure A.3.** Data dictionary - labels from the IKNL dataset. The complete dictionary is available at the website of the Cancer Registry of Netherland ([16]).

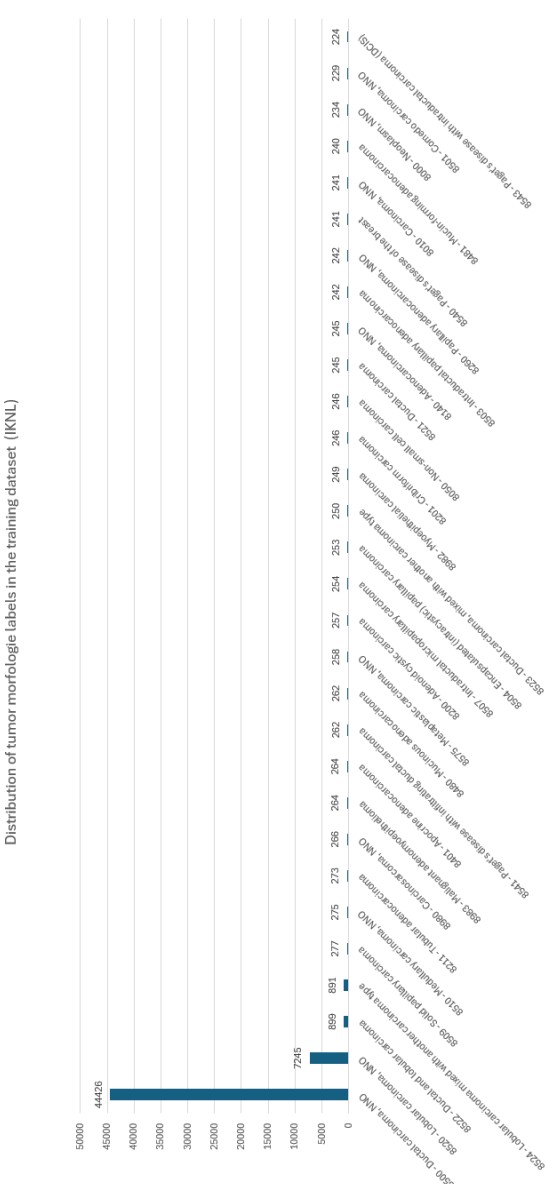

**Figure A.4.** Distribution of tumor morphology in the IKNL dataset (the training dataset for this experiment).

