# OpenReview forum: "Keeping it Simple – Computational Resources in Deep Generative versus Traditional Methods for Synthetic Tabular Data Generation in Healthcare"
_NLDL.org/2025/Conference — Submitted to NLDL 2025_

### Official Review · Reviewer_6F7G · 2024-09-23
**Nice idea, limited execution, muddled presentation**

**Confidence:** 4

**Summary:**

This paper performs an experiment on synthetic healthcare data, comparing a simple statistical model (Gaussian copula) with two more sophisticated machine learning methods (a generative adversarial network and a variational autoencoder, both designed for tabular data).

The main result of the paper is that on a synthetic dataset mimicking measurements of breast cancer patients, the Gaussian copula generates synthetic data which is of comparable quality to the other methods while requiring far less computational resource.

The effect of including or omitting metadata, which ties different tumours to the same patient, is also investigated. The effect of omitting metadata on runtime is dramatic. The performance of all methods decreases in the absence of metadata, but the copula method is still competitive.

**Strengths:**

Coherence and correctness: The basic message of this paper is an important one, and one which has appeared occasionally in other areas of the machine learning literature: the performance of sophisticated machine learning methods is often matched or outperformed by simple statistical models. Those simple models require far fewer computational resources, having important downstream impact (e.g. environmental impact). This is an important message that needs to sink in to the machine learning community.

Clarity and presentation: The main message of the paper is well described. The structure of the paper is fine. The introduction places the experiment in context, and there is a good discussion of limitations and potential for future research.

There are some insights into how the presence or absence of metadata influences output, which I hadn't encountered before. Actually, I would like to see this discussed a bit more. Can you speculate on the mechanism by which metadata influences performance?

**Weaknesses:**

1. Incrementality: The scope of the experiment described is quite limited, making it very difficult to gauge its generalizability. There are no measures of uncertainty given.

(a) The experiment is performed on only one dataset. What should we expect if we were to use a different dataset? I appreciate that it is infeasible to use real data here, but what about other synthetic data? As a minimum there should be some sort of cross-validation to investigate robustness to the training/test split within the dataset used.

(b) There is no exploration of the effect of hyperparameters. It is stated that this is "To streamline and facilitate ... reproducibility" (line 206) but I am not convinced by this. It should not be beyond the wit of a reader to reproduce the experiment with a different set of hyperparameters.

(c) Three methods are compared. I think this is a bit unambitious, particularly since one of them, TVAE, was introduced by its authors partly to demonstrate its **inferiority** compared to their primary method from the same paper. For example, diffusion models are mentioned (line 036) as another method for simulating datasets, and there do exist diffusion models for tabular data - see for example TabDDM (ICML, PMLR 202:17564-17579, 2023). How does this compare?

(d) The study does not address the presence of missing data, which was removed.

2. Clarity and presentation: The description of the methods of this paper are not given in sufficient detail.

(a) The summary the main methods under study (170-184), Gaussian copulas, CTGAN, and TVAE, are too brief. The description of TVAE only gets a single sentence, but I think it would be important for a reader to have a bit more insight into its mechanism. I understand that there is a page limit, but then the fact that their pipeline removes 10 rows of missing data was granted nine lines, so there should be room to tighten the exposition. There should be room for some brief mathematical description, which is entirely lacking in this paper, or even a fuller exposition in a much-expanded Appendix.

(b) It was not clear to me what the role of the test dataset is here. It is mentioned on line 198 as a way to "compare the performance", but then not mentioned again. The quality evaluation talks about similarities with the training data but not test data. For a problem like synthetic data generation, it is important to make a clear conceptual distinction between the main problem at hand (unsupervised) and methods of performance evaluation ('supervised'). The supervised part is lacking in description. Compare ref 10 which has the completely transparent statement: "When training a classifier or a regressor to predict one column using other columns as features, can such classifier or regressor learned from T_syn achieve a similar performance on T_test, as a model learned on T_train?" A statement along those lines (and follow-up details) is needed here.

(c) It would be helpful to give a mathematical description of the quality scores, either in the main text briefly or more fully in an Appendix. The description in 227-238 does not make it clear enough which scores are suitable for which types of data. Clearly distinguishing 'categorical' and 'real-valued' data would help. (Descriptor like 'numerical' are 'real' are unclear to me.)

3. Clarity: The standard of English is adequate, but it could be improved. There are quite a few clunky phrases - I have picked out a few where the choice of words might be making the meaning actively misleading.

065 - "benchmarks ... with deep generative methods" should be "benchmarks ... against deep generative methods" ? Line 088 similarly.

082 - "enhancement of GANs". I am not sure what this means. "enhancement given by the use of GANs"?

101 - "the experiment" -> "an experiment"

124 - "on the are" -> "on the data are" ?

206 - Sentence starting "To streamline" is uninterpretable as the subject is missing its verb.

4. Presentation: Figures are not well presented. Figures 1 and 2 are too small. It took me a while to work out why two different scoring mechanisms, Range and Category, have been interspersed with each other, before I realised that one is for real-valued and one is for categorical data. (See above.) Figure A.3 is mostly in Dutch but the "English" column, I imagine, is supposed to be an English translation. This is needed to be able to interpret the codings.

5. Reproducibility: I did not see any evidence that the code used to generate the experiments has been made available. Will this change?

**Final Rebuttal Confidence:**

4

**Final Rebuttal Justification:**

The authors have taken some steps towards addressing my and the other reviewers' concerns, and have done what can reasonably be expected given the timeframe. There are modest improvements to the presentation and some expanded explanations. But there is still essentially no mathematical exposition, and the primary weaknesses, identified by more than one reviewer, remain, particularly that both the choice of comparator methods and the scope of the experiments is too narrow. No new experiments have been performed, and the request for a comparison with diffusion models is left for future work.

If the scoring system allowed for fine-grained recommendations I might raise the score slightly, but since I can choose only between accept and reject I feel this paper still clearly falls in the latter category.

**Justification:**

This is an interesting idea for a paper with an important central message, but its execution is too limited in scope. I would like to see a more ambitious experiment, including some or all of: inclusion of other methods, estimation of uncertainty, some exploration of relevant parameters. A secondary issue is the main description of the work which is too vague and lacks mathematical precision. Together with a lack of code, this makes reproducibility an issue.

---

> ### Author Rebuttal · Authors · 2024-10-24
>
> We sincerely appreciate the thorough review and insightful comments. We believe that the revisions made based on your suggestions have significantly improved the quality of the paper.
> Regarding the incrementality of our study, we acknowledge that our experimental scope was constrained by time-limited access to High-Performance Computing resources, and therefore had to limit the investigations. Nevertheless, we have taken steps to address this limitation as follows:
> We have updated the "Limitations" section of the manuscript acknowledging lacking robustness of generalizations from the conclusions due to only one dataset being used, a limited number of generators and using default hyperparameters.
> For future work, we have updated the text according to your suggestions:
> “To investigate the robustness of the conclusions, there is a need for a wide range of benchmarks on a range of different datasets, investigating other types of both deep and traditional generators and for differing use cases and data modalities, including more thorough hyperparameter exploration. (…) In addition, future research should investigate the differences in quality scores for a broader range of evaluation criteria covering topics like fairness, privacy and other representativity (similarity and usability) metrics.“ We also plan for future research on testing both diffusion models and LLMs.
> Regarding the missing data, we address in the manuscript that removing rows with missing data is not optimal for an analysis in a real-life setting. In healthcare datasets, missing values in themselves can carry meaningful clinical information. We acknowledge this limitation in the "Data preparation" section, stating that "in a real-life experiment, this may not be an optimal approach as missing data fields is an inherent characteristic of healthcare data."
> We also appreciate the reviewer’s detailed feedback clarity and presentation of our methods, that helped us update and clarify where needed. We have addressed the points raised as follows:
> The descriptions of the models (GC, TVAE and CTGAN) have been expanded in the "Choice of Synthetic Data Generation Models" section, providing more detail. Regarding the use of the test dataset, it should have been referred to as such fin the performance evaluation and not as the training data, thank you for pointing out this error. This has been updated and clarified in the revised manuscript. We have also supplied specific references to the SDV library's "Quality Report" and "Diagnostic Report" documentation, which contain deeper explanations of the methods used and associated visualizations.
> As non-native English speakers, we are grateful for your helpful comments on typos and grammatical issues., these suggestions were invaluable during our revisions (clarity).
> Presentation: We have added larger versions of the figures in the appendix to improve readability, as we could not find how else the template allowed larger figures. We appreciate you pointing out the lack of clarity on "Range and Category" and have updated the text to clarify this point. Additionally, we have revised the coding in figure A3 with a proper English translation as requested.
> Reproducibility: While the entirety of the code is not publicly available at present, we have ensured reproducibility through: (1) the use of publicly available data; (2) the open-source SDV library; (3) the description of generator classes from SDV in the manuscript; and (4) the application of default parameters for all functions. We believe that these measures provide sufficient detail for accurate replication of our results. We are considering making the code publicly available, in line with our commitment to open science. In the meantime, we remain open to providing additional details as needed to support reproducibility efforts.

---

### Official Review · Reviewer_2fBg · 2024-10-06
**The initial review**

**Confidence:** 3

**Summary:**

This paper presents a comparative study that evaluates computational resource consumption and data quality in generating synthetic tabular data. The authors test the traditional Gaussian Copula method and popular deep generative methods, such as TVAE and CTGAN, using the Dutch cancer registry dataset. The comparisons indicate that the traditional method requires fewer computational resources, while the quality of the generated data is comparable to those from generative AI models.

**Strengths:**

- This paper investigates an interesting problem: do we really need computationally-intensive generative models for some specific tasks?

- The authors provide detailed information on the experiment setups, evaluation metrics, and result analysis.

- The authors discuss the limitations and future work related to the proposed evaluation metrics.

**Weaknesses:**

- The experimental setup should be coherent, but some sections contain repetitive content. For example, the hardware setup is discussed twice. This section could be made more concise.

- To enhance the clarity of the paper, it would be beneficial to briefly describe the deep generative models utilized in the experiments, similar to the descriptions provided for the traditional model.

- I also have some questions listed below:

    - For Table 2, why is the CTGAN model not evaluated under the "without meta data" scenario?

    - For quality evaluation metrics, why is the SDV library particularly being used? Are there any other measurements that can be included?

**Final Rebuttal Confidence:**

2

**Final Rebuttal Justification:**

I think the authors have addressed my initial concerns. The pros are that the authors have presented their methods, experiments, and results reasonably well for this specific topic. The concerns are that it seems many of the other reviewers are skeptical about the novelty and the research scope of this paper after reading their comments. I have lowered my confidence due to my limited knowledge of tabular data generation or healthcare data.

**Justification:**

I think this paper has shown a good presentation of the experimental setups, evaluation metrics, and results analysis, which indicates that the authors have conducted a detailed and systematic empirical analysis. However, this paper can also benefit from some refinements, especially in paper organization and clarity. Those would further improve the paper quality.

---

> ### Author Rebuttal · Authors · 2024-10-24
>
> Thank you for your detailed and helpful comments! We truly appreciate the improvement suggestions regarding repetitive content, which were very helpful as we revised the text (particularly in sections “3.4 Setting up the infrastructure” and “4.1 Pipeline requirements for processing capacity (Q1)”).
> We also appreciate your recommendation to better describe the deep generative models utilized. In response, we have revised the text in "3.3 Choice of synthetic data generation models" to strengthen the manuscript as suggested.
> Your question regarding the lacking evaluation of CTGAN without metadata highlighted an important oversight in our initial reporting which supports the conclusion on computational resource need for some deep generative models. Due to CUDA out-of-memory errors in our HPC environment, we were actually unable to complete the CTGAN evaluation under the scenario "without predefined metadata". This technical limitation should have been documented in our original submission, and we have now corrected this. Specifically, we have (1) added an explicit explanation in the text, (2) updated Table 1 to reflect the missing CTGAN results with proper notation, and (3) included a footnote in Table 2 explaining the CUDA memory constraints.
> Regarding our choice of the Synthetic Data Vault (SDV) library, our decision was based on several key factors: (1) its intuitive design and well-documented structure, which facilitates reproducibility; (2) its citation in the field, indicating reliability and acceptance; and (3) its origins as a project from MIT, lending academic credibility. We have added this reasoning to the Methods chapter.  We acknowledge that relying on a single library may be viewed as a limitation. In response to your feedback, we have addressed this in the "Limitations" section of the manuscript and expanded our recommendations for future work. We suggest that future studies explore alternative libraries alongside SDV, such as SynthCity, an open-source library developed at Cambridge University, which we highlight as a potential candidate for future comparisons.

---

### Official Review · Reviewer_rJLi · 2024-10-14
**The paper lacks novelty and suffers from weak experimental design, insufficient analysis, and limited practical relevance.**

**Confidence:** 4

**Summary:**

The paper explores the computational resource requirements and performance of deep generative models (such as CTGAN and TVAE) compared to traditional methods for generating synthetic tabular healthcare data. The authors benchmark the methods using a synthetic breast cancer dataset and provide an analysis of computational resource requirements, and performance on data similarity.

**Strengths:**

1. The paper discussed an important topic related to the computational costs of various synthetic data generation methods.
2.  Choosing different hardware setup (Laptop, Azure) gives good insights about running the models in different setups.

**Weaknesses:**

1. Lack of Originality and Novelty: The paper does not present any new concepts, methodologies, or substantial contributions to the field. The paper claims to explore computational resource requirements, but the results are predictable, confirming the well-known fact that deep generative models are resource-intensive.
2. The paper positions itself as a comparative study, but there’s no evidence that the study significantly builds on the existing work, espically that the related work section is very limited.
3. As mentioned in the limitations section of the paper, the results of the paper depend heavily on synthetic dataset (from the Dutch Cancer Registry), which undermine the validity and generalizability of the findings. In healthcare, it is crucial to assess how these methods perform on real-world data, which contains noise, missing values, and variability that synthetic data often lacks. A robust comparative study requires multiple datasets with different characteristics (e.g., size, feature types, distributions) to demonstrate the generalizability of the findings.
4. The paper compares only limited deep generative models (CTGAN and TVAE) against a single traditional method (Gaussian Copula). These models are not necessarily the best or most representative of their respective classes. For instance, newer and potentially more efficient models (such as diffusion models) should be discussed. Additionally, the choice of methods is not adequately justified. Why these specific models were chosen remains unclear.
5. No Real Statistical Comparisons: The analysis fails to include proper statistical testing or comparisons of the performance results. Simple metrics are reported (e.g., column shape similarity), but there’s no statistical comparison (e.g., t-tests, etc) to validate whether the differences between the models are statistically significant. The authors draw conclusions based on these results without showing whether the differences observed are meaningful.
6. Over-Simplification of Results: The results section lacks depth and detail. For example, when comparing the models on their ability to generate accurate data, the authors do not explore why certain models perform better or worse for specific types of data (e.g., numerical vs. categorical). The results are mostly reported in aggregate, masking any deeper insights that could be gained from a more granular analysis.

**Justification:**

The primary reason for rejecting this paper stems from its lack of novelty and insufficient contribution to the fild. The study fails to introduce any novel methodologies, frameworks, or improvements.

---

> ### Author Rebuttal · Authors · 2024-10-24
>
> Thank you very much for your thoughtful review and insightful comments. You are indeed correct in noting that the results are not entirely surprising. In the literature, the significant resource consumption of deep generative models is often taken as a given. However, during our review of existing comparative studies documenting the extent of the differences in complexity, cost, and effort between traditional and deep generative methods, we found that there was limited coverage of this aspect. We, therefore, consider our study to make a valuable contribution in starting to fill this gap.
> Our intention was to perform a comparative study as in “comparing two or more cases with quantitative methods to detect similarities or differences”. We will make sure to clarify this point in the revised manuscript and will update the Related Work section to highlight the existing gap of documentation regarding computational differences.
> In our updated manuscript, we have included your suggestion that the experiment should ideally be repeated using different datasets, other generative models, and a broader set of evaluation criteria in the section on further research. This is an excellent point, and we appreciate your input.
> Regarding your observation on how many synthetic datasets can be unrealistic, this is a very valid point and probably true also for many real open datasets. The story of this particular dataset is actually a bit interesting in that regard. It was created by the Dutch National Cancer Registry (NCR) with the intention to let the synthetic dataset inherit the typical traits of real-world healthcare data – meaning both the structure and statistical patterns but also missing values etc. to ensure researchers and developers will get a better understanding of the nature of healthcare data before they apply for access to the real data. On their website, they have emphasized that the dataset “gives a good impression of the data available in the NCR for researchers who wish to apply for data in the NCR”.
> We chose the three models based on their well-established status, thorough documentation, and availability through the MIT-developed SDV library. Thank you for pointing out that this was not sufficiently explained in the original version of the paper. We have now included this clarification.
> We agree with your suggestion to explore newer and more advanced models, such as diffusion models or LLMs, and plan to address these in future research.
> Since our intention was to keep the text concise and focused on the comparison of computational resources, we relied on two pre-configured evaluation reports. These were provided by the SDV library: “The Diagnostic Report” and “The Quality Report.” The reports offer a statistical comparison of the original and synthetic datasets through aggregated scores. It is true that average scores do not provide the necessary detail for deeper insights into all aspects of quality. In the revised manuscript, we have highlighted this limitation and added some nuance to details in the statistical comparisons (differences between numerical and categorical values, and discussion on why the GC completely failed for the variable tumor morphology).
> Our focus in this project was to document the exact differences in environmental footprint for comparable similarity within certain applications. We agree with your perspective that, to fully address the question of how the models differ across all aspects, a more comprehensive evaluation is needed. This would include not only statistical similarity in terms of coverage and diversity but also aspects such as downstream clinical usability, privacy metrics including shadow modelling attack simulations, and fairness metrics. We recognize that these differences may vary depending on the intended downstream use of the models. In response to your suggestion, we have clarified the limitations in this experiment and included an aspiration to conduct broader quality evaluations in the future research section, along with plans to explore additional models and datasets.

---

### Official Review · Reviewer_42fL · 2024-10-15
**Review of Keeping it Simple – Computational Resources in Deep Gener- ative versus Traditional Methods for Synthetic Tabular Data Generation in Healthcare**

**Confidence:** 5

**Summary:**

The purpose of the study is aimed at supporting the broader healthcare community and challenges faced with safe data access with the fast paced growth of AI tool development. Particularly, how generative AI methods can be used to engage synthetic data with realistic results and low identification risk compared with traditional tabular methods or traditional statistical methods of research. The study hopes to include more research into the needs of benchmarking and computational needs for tubular data in healthcare. The researcher undertakes a trial experiment with a synthetic generated dataset from the Dutch cancer registry of 60,000 hypothetical breast cancer patients. The experiment compares the performance of traditional statistical methods and deep generative methods both on similarity and computational resources to guide the practical use of these tools to investigate the following questions:

Q1: What are the practical resource requirements for running the pipelines of deep generative methods for generating tabular data compared to traditional generators, and
Q2: How well does deep generative methods perform for generating tabular data compared to traditional generators in terms of statistical similarity?

The traditional statistical method undertaken in the study is Gaussian Copula (Synthesizer), which is considered the benchmark. This model was used to compare the two generative AI models. The researcher pairs the experiment with the Conditional Tabular Generative Adversarial Network (CTGAN) and Tabular Variational Autoencoder (TVAE). The generative AI is run both on a laptop and virtual machine to explore questions of sustainability (with the laptop being the sustainable approach). The Synthetic Data Vault (SDV) pipeline  for evaluation metrics for the models were compared to answer Q2, and practical performance measures (size and runtime) was gathered for answering Q1.

I will discuss data preparation, pipeline requirements and quality evaluation across weaknesses and strengths. The overall outcome was that the deep generative models had potential to run effectively with tabular data but only when the meta data was not defined, yet in the actual experiment the results were less favourable to generative AI than traditional methods. The TVAE outperformed the Gaussian Copula when meta data was not available but when the meta data was available, the results were not available. The CTGAN had to be aborted due to resource overload after two days. The study outlines that for this type of experiment the right choice of model will depend on the nature of the data in the different healthcare scenarios. All pipelines discussed were faster when meta data was manually defined. The study acknowledges it is a synthetic study with a synthetic dataset that can only be used in the context of a methodological test. More research and trialling would be required to develop more succinct evaluation metrics.

**Strengths:**

The strengths of the study is that the research does complete part of it's goal to address new areas of research in comparing traditional and deep generative methods in terms of computational resources needed, relative to differences in statistical similarity between the training dataset and the synthetic dataset. The study reveals that while deep generative approaches can produce results of similar (or in some cases with the TVAE) higher results, it comes at the cost of High Computing Resources which impact sustainability concerns related to AI in general. The higher TVAE results were only gained when the meta data was not available which the study acknowledges is not the most optimal approach for healthcare studies.

With a data split of 20/80 (with 20% used to train the generative AI models, 80% used for testing; to compare performance and data) the tests were done. The results relating to the speed of the information gained were interesting. For the test without meta data defined, the traditional GaussianCopula model used 17 seconds, the TVAE used 1200 seconds and the CTGAN used 9600 seconds on the Azure server and without meta data. With meta data the training was faster for all models, rendering the relative differences smaller but with a similar distribution. Storage was notably smaller when the meta data was defined.

A strength of the research is the capacity to report on the results of the generative AI versus traditional statistical methods with tabular data. With meta data, the TVAE performs at a sufficient level, but does not perform better than the simple statistical approach (Gaussian Copula) on the tabular data, with slightly lower scores for column shapes, column pair trends and coverage. The TVAE shows slightly lower quality scores than the CTGAN and lower coverage. When meta data was not manually defined, the
TVAE showed better overall similarity performance scores than the Gaussian Copula. The study also acknowledged that using the Dell laptop was possible on the TVAE generator but took so long it was not practical. This option was still faster than the CTGAN that was stopped for practical reasons after two days. Both deep generative methods required Higher Computing Resources which are not supportive of sustainable research outcomes, and this is a positive result for the study to support the larger framework of research area around computational needs.

**Weaknesses:**

The study is open that it used the SDV Evaluation Metrics Library from the original Synthetic Data Vault Project. It was deemed sufficient  when using this pipeline (along with the data preparation) to use minimal bias or fairness metrics. The study acknowledges that this is beneficial for benchmarks but the outcome of this choice can obscure deviating performance for specific diagnostic features. The question is pertinent that any research on generative models of AI, particularly in the healthcare system demonstrate a significant attempt to minimise the neglect of bias and crucial neglect in downstream models. While the synthetic data was a sample, and not intended for real life consumption, the reality is that the medical and healthcare industry are rampant with bias against marginalised bodies and this impacts diagnosis statistics and information. The test data chosen was breast cancer data, which is predominantly women, which falls within a the category of marginalised body. The study acknowledges that when choosing a generative method, researchers need to consider which quality dimensions are the most practical and lists sustainability, usability, privacy, bias and fairness as factors in terms of impacting the downstream data set. But in evaluation benchmarking with generative AI, what is the true purpose of studies that evaluate a system that continue to perpetuate systems of oppression, as opposed to engaging with new approaches from the outset?

A weakness of the study is that the results of of the CTGAN were not pursued, as GAN's (unlike their descriptive counterparts in generative models) are designed to learn the underlying probability of distribution in data. This is not to say they are not without limitations (mode collapse and non convergence / instability) but research specifically into healthcare synthetic data supports the use of GAN based synthetic data that enables the generator to learn a fair deterministic transformation based on a well-defined notion of algorithmic fairness. This mode of thought and process is important for healthcare research to support healthcare organizations to improve care delivery in the era of value-based healthcare, digital innovation, and big data. While the logic behind the GAN not being pursued was time management and sustainability factors which are reasonable, is it not the responsibility of the greater AI and research community to ensure that all research dedicated to generative AI (in particular areas that are predisposed to bias amplification) provide significant frameworks to ensure research addresses these risks as a core feature of their study and research?

**Justification:**

The study has been successful in achieving it's goal to explore the outcomes of TVAE generative AI against traditional statistical methods Gaussian Copula. But was not successful in obtaining clear results for the CTGAN generative AI due to time limitations. The results support research into sustainable methods of traditional and deep generative AI and consider factors such as High Performance Computer resources, sustainability and the differences the approaches have with between the training dataset and synthetic dataset.

The study states that as it is a methodological test, the objective quality was not truly determined and the researchers acknowledge that the study should be completed with a real dataset and for a defined downstream application. Other quality metrics and mechanisms should also be added, to evaluate the clinical us- ability and logic of the resulting synthetic data. I agree with this evaluation as the existence of bias in algorithmic forms, which can be cognitive bias, social bias, statistical bias or any other sort of bias, can result in an inaccuracy which is systematically incorrect. Within AI bias research, there is a misconception that a bigger dataset will provide more accurate results. This belief has held because one can usually increase the accuracy by adding more data. But unfortunately, this increased accuracy won’t translate to in-production accuracy if the additional data is biased and not reflective of the real world.

Data scientists and researchers need to be comfortable with the concept that algorithms in general are predisposed to bias (more so descriptive AI than generative) and engage processes that optimise their data to minimize bias amplification. This is specifically imperative in areas where bias is a known problem (ie healthcare.) While this study has chosen to use generative AI methods, which are considered one step to minimise bias amplification, more steps in the data preparation could be taken to ensure that AI is being used an opportunity for social equity. Particularly if the trial data relates largely to marginalised bodies. There are significant studies that discuss the use of AI creating opportunities for accurate, objective and immediate decision support in healthcare with little expert input specially valuable in resource-poor settings where there is shortage of specialist care. Given that AI poorly generalises to cohorts outside those whose data was used to train and validate the algorithms, populations in data-rich regions stand to benefit substantially more vs data-poor regions, entrenching existing healthcare disparities. These considerations need to be taken into account for all studies to be truly of value to the greater healthcare community and larger impact of social equity.

---

> ### Author Rebuttal · Authors · 2024-10-24
>
> Thank you for your engaging comments! Our choices of models and evaluation criteria do as you say reflect that the objective of the experiment is to showcase computational complexity and environmental concerns, and so we chose to not include bias and fairness metrics at this point.
> The CTGAN was not evaluated under the scenario without meta data since we were actually unable to complete the CTGAN evaluation due to CUDA out of memory errors in our HPC environment. We have made this technical limitation more clearly documented in the paper submission. This result in itself actually highlights the conclusions on resource needs which was the focus for this research. We have also detailed and added more nuance in the discussions on the differences in quality, in performance for numeric vs categorical variables and the fact that the TVAE had a more robust performance independently of whether the meta data was defined.
> We absolutely agree with your arguments in that bias is an important topic in evaluation of data driven methods, particularly in high-impact fields like healthcare. We also believe that evaluation criteria should be chosen based on the context of the task at hand and the intended downstream application. We did not include fairness benchmarks in this study due to the nature of the dataset, and as it is based on a quite homogeneous population.
> We have made these arguments clearer in the discussion section and the further research section to emphasize that evaluating bias and fairness is of high importance in further research.  Our initial search did not provide any good references documenting the difference in bias propagation in traditional vs generative methods but we would love to include any if you have some tips.

---

### Meta-Review · Area_Chair_oV3g · 2024-11-01

**Recommendation:** Reject
**Confidence:** 5

**Metareview:**

This paper addresses an important yet well-known concern regarding the computational resource demands of deep generative models versus traditional statistical approaches for synthetic healthcare data. While the study highlights the efficiency of Gaussian copula models in comparison to deep generative models, the experimental scope is notably narrow. The lack of newer generative models, broader datasets, and thorough statistical testing limits the generalisability and impact of the findings. The paper’s insights into metadata handling and resource constraints in healthcare AI are valuable, yet more comprehensive exploration is needed.

Despite improvements in response to feedback, the paper falls short of advancing novel methodologies or yielding broadly applicable insights.

**Suggested Changes To The Recommendation:**

3: I agree that the recommendation could be moved up

---

### Decision · Program_Chairs · 2024-11-06

Reject